# Feature Pyramid Networks and Long Short-Term Memory for EEG Feature Map-Based Emotion Recognition

**DOI:** 10.3390/s23031622

**Published:** 2023-02-02

**Authors:** Xiaodan Zhang, Yige Li, Jinxiang Du, Rui Zhao, Kemeng Xu, Lu Zhang, Yichong She

**Affiliations:** 1School of Electronics and Information, Xi’an Polytechnic University, Xi’an 710060, China; 2School of Life Sciences, Xidian University, Xi’an 710126, China

**Keywords:** biological signal processing, emotion recognition, EEG feature map, feature pyramid networks, long short-term memory

## Abstract

The original EEG data collected are the 1D sequence, which ignores spatial topology information; Feature Pyramid Networks (FPN) is better at small dimension target detection and insufficient feature extraction in the scale transformation than CNN. We propose a method of FPN and Long Short-Term Memory (FPN-LSTM) for EEG feature map-based emotion recognition. According to the spatial arrangement of brain electrodes, the Azimuth Equidistant Projection (AEP) is employed to generate the 2D EEG map, which preserves the spatial topology information; then, the average power, variance power, and standard deviation power of three frequency bands (α, β, and γ) are extracted as the feature data for the EEG feature map. BiCubic interpolation is employed to interpolate the blank pixel among the electrodes; the three frequency bands EEG feature maps are used as the G, R, and B channels to generate EEG feature maps. Then, we put forward the idea of distributing the weight proportion for channels, assign large weight to strong emotion correlation channels (AF3, F3, F7, FC5, and T7), and assign small weight to the others; the proposed FPN-LSTM is used on EEG feature maps for emotion recognition. The experiment results show that the proposed method can achieve Value and Arousal recognition rates of 90.05% and 90.84%, respectively.

## 1. Introduction

Emotion is a state that integrates a variety of complex feelings, thoughts, and behaviors of people. It includes the psychological response to external or self-stimulation, as well as the physiological response accompanying the psychological response [1]. EEG signals of people are directly generated by the nerve center. The nerve center is closely related to mood change and can better express the subtle changes of the emotional state. It can not only directly reflect the mood state of people, but also respond to the characteristics of real-time emotional changes [2,3]. The emotion recognition research based on EEG has become one of the topics of research that focuses on human–computer interaction.

In recent years, with the rapid development of deep learning algorithms, more and more scholars have applied them to the field of EEG emotion recognition, with some achievements [4,5]. Huang E et al. proposed the Dual-Stream Convolutional Neural Network to use the extracted time-domain features and frequency-domain features for linear weighted fusion for classification training, which improved the classification accuracy of subjects [6]. Li Chang et al. applied multitask learning in deep learning technology to EEG emotion recognition and verified the effectiveness of their methods on a DEAP dataset [7,8]. The independent subject emotion recognition algorithm of the Dynamic Empirical Convolutional Neural Network was proposed by Liu Shuaiqi et al., and combines the advantages of empirical mode decomposition and differential entropy. The accuracy of this algorithm is 3.53% higher than the existing emotion recognition methods [9]. Jiang HP et al. proposed the CSP_VAR_CNN (CVC) emotion recognition system, which is based on the convolutional neural network (CNN) algorithm to classify emotions of EEG signals; the average accuracy reaches 69.84%, which is 0.79% higher than that of the CVS system [10]. Zhang et al. thresholded the continuous emotional trajectories, and then classified emotions through the emotional classification framework of long short-term memory networks, which improved the classification accuracy [11]. Song TF et al. used Dynamical Graph Convolutional Neural Networks for emotion recognition on SEED and DREAMER databases, in which the average recognition accuracy was 90.4% on the SEED database, and 86.23% and 84.54% for Valence and Arousal classifications on the DREAMER database, respectively [12]. Chakravarthi et al. proposed automated CNN-LSTM with ResNet-152 algorithm; its recognition accuracy on a SEED-V EEG dataset reached 98% [13]. 

When collecting EEG signals, all electrodes are not on the same plane, and each channel is in a different spatial position [14]. Therefore, EEG signals contain not only time-domain information but also spatial topology information between each channel. This spatial information corresponds to different subarea functions of the brain, which has specific significance and is closely related to the final EEG signal category [15,16]. However, research shows that the original EEG signal can be spatially converted by certain methods to retain its spatial topology information. Li et al. designed a hybrid model, which used CNN to extract the spatial feature information of the EEG signal, and then inputed it into RNN to extract EEG timing information. The accuracy of emotion recognition was 72.06% and 74.12%, respectively [17].

Based on the above research, the work of this paper was as follows:

(1) To solve the problem that the original EEG cannot retain spatial information, we proposed an EEG feature map generation method based on AEP, which can retain the spatial topological information of EEG. It transformed a 1D data sequence to a 2D color image, which had a more intuitive visual effect.

(2) To solve the problem of data redundancy, we only used three frequency bands (α, β, and γ) of data, distributed the weight proportion for channels, assigned large weight to strong emotion correlation channels (AF3, F3, F7, FC5, and T7), and assigned small weight to the others.

(3) We proposed to use FPN-LSTM on the EEG feature map. FPN was better at small dimension target detection and insufficient feature extraction in the scale transformation. Compared with LSTM, CNN, KNN, and CNN-LSTM, FPN-LSTM obtained better recognition results based on the experiments of this paper.

## 2. Materials and Methods

### 2.1. Generation of EEG Feature Map Based on AEP

#### 2.1.1. EEG Feature Selection

A large number of studies show that the electrical signal activity of cortical neurons is relatively weak. In order to analyze the characteristics of EEG signals, the FFT method is used to convert EEG signals to a frequency domain. EEG signals are divided into five basic rhythms δ (0–4 Hz), θ (5–8 Hz), α (9–13 Hz), β (14–30 Hz), and γ (≥31 Hz) according to different frequency bands. Different rhythms represent different states of each person; the representative characteristics of the five basic rhythms are shown in Table 1.

It can be seen from Table 1 that the three rhythms of α (9–13 Hz), β (14–30 Hz), and γ (≥31 Hz) are most closely related to emotion, so in this paper, the three rhythms are selected as the frequency bands of the EEG feature maps, and the average power, variance power, and standard deviation power are used as the feature samples of each frequency band, so as to build the EEG feature map dataset.

#### 2.1.2. EEG Feature Map

The placement position of electrodes for collecting EEG signals according to the international 10–20 standard is shown in Figure 1, where Figure 1a is the distribution map of skull top electrodes, Figure 1b is the distribution map of skull side electronics, and Figure 1c is the electrode distribution plan of 64 active AgCl electrodes according to the International 10–20 System, and among this the red contents are 32 electrode distribution. In this paper, the sampling frequency is 512 Hz, and the red electrode is 32 channels of collected electrodes.

Considering the spatial distribution of electrode positions and combining with the International 10–20 system 64-channel electrode distribution map, the three-dimensional spatial data of EEG were converted into two-dimensional image data, while retaining the spatial topology information of EEG signals. In this paper, the AEP method was used to transform a 1D EEG signal into a 2D EEG feature map, which was from a spherical coordinate system to a Cartesian coordinate system. The human head is a sphere-like structure, which is called the natural reference surface. AEP eliminates map distortion caused by a natural reference plane and maps each position of the brain surface to the corresponding position in the projection plane. Another definition of AEP is the mathematical transformation equation that converts 3D electrode coordinates to 2D grid coordinates. The real spatial position was mapped to a 2D plane through projection, and the original features of the spatial position were retained. The specific implementation is as follows:

Step 1, the Cartesian coordinate system coordinates of each electrode channel in 3D space are (x,y,z), and then transforming (x,y,z) to spherical coordinates (polar coordinates), obtaining the parameters of r, e, and a, the equations are shown as follows:(1)$$r=\sqrt{x^{2}+y^{2}+z^{2}}$$
(2)$$e=\arctan\frac{z}{\sqrt{x^{2}+y^{2}}}$$
(3)$$a=\arctan\frac{y}{x}$$
in which, r is the radius of the point in polar coordinates after projection, e is the elevation, a is the azimuth.

Step 2, converting the polar coordinates obtained into rectangular coordinates (X,Y):(4)$$X=\left(r\times \tan e\right)\times \cos a$$
(5)$$Y=\left(r\times \tan e\right)\times \sin a$$

Step 3, calculating the position information of each coordinate system of all acquisition channels according to the above steps, and drawing the distribution diagram of channels in 2D Cartesian coordinate system. 

After the electrode distribution position of the 2D EEG matrix map was obtained by the above method, the eigenvalues of each subject of α, β, and γ frequency bands were taken as the data of the EEG matrix map and the blank pixels among the electrodes were completed by the Bicubic interpolation algorithm. Then, the EEG matrix maps of three frequency bands were used as the data of the R, G, and B channels for the EEG feature map, so as to obtain the EEG feature color image with spatial information. This image is shown in Figure 2. 

### 2.2. FPN-LSTM Feature Extraction Network

FPN is a feature extractor designed according to the concept of a feature pyramid, which is a feature enhancement network. It aims at improving the feature extraction method of CNN, so that the output features can better represent the feature information of each dimension of the input image. The FPN has three basic processes:

(1) The bottom-up path, namely, the bottom-up generation of features of different dimensions, is used to construct a higher-resolution layer from a semantic-rich layer. The layer constructed in this way has high resolution, rich semantic, and repeated up-sampling and down-sampling. Therefore, a lateral connection is constructed between the reconstruction layer and the corresponding feature map, so that the detector can better predict the location.

(2) The top-down path, that is, top-down feature complement enhancement. Upsampling (interpolation method) is used, i.e., on the basis of the original image pixels, the appropriate interpolation algorithm is used to insert new elements among pixels, so as to expand the size of the original image. By analyzing the feature map, the size of the feature map sampled is the same as that of the next layer.

(3) Lateral connection, which fuses the upsampling results with the bottom-up generated feature map, is the correlation expression between the network layer features and the final output features of each dimension.

In this paper, in order to solve the problem that CNN’s translation of the target remains unchanged and cannot adapt to scale transformation, there is a problem of insufficient feature extraction during scale transformation. Using FPN to improve CNN-LSTM, the comparison process between FPN-LSTM and CNN-LSTM is shown in Figure 3.

Figure 3 is divided into two parts. One part is the original network CNN-LSTM used in this paper, which is composed of three convolution layers with convolution kernel size of 3 × 3, three ReLU function layers, three maximum pooling layers, LSTM, a full connection layer, and Softmax classifier. The other part is the FPN-LSTM network proposed in this paper. The processing process of EEG feature map is as follows:

First, the EEG feature map of 16 × 16 × 3 was transformed into 8 × 8 × 16 after two times of 5 × 5 convolution, and then 3 × 3 convolution was used to transform it into 6 × 6 × 32. Second, 1 × 1 convolution was used to transform the number of channels to be fused into 128, and the EEG feature maps of different scales were fused by concat. Then, a 5 × 5 pooled kernel was used to transform the feature map into 2 × 2 × 128. Finally, the features were entered into the LSTM network to extract the features of EEG timing information, and the classification results were output by Softmax. 

It can be seen from Figure 3 that the FPN-LSTM network model integrates deep features and shallow features, and improves the model feature extraction capability. Compared with the CNN-LSTM model, it integrated the features of different scales and enhanced the spatial topology information better. Therefore, FPN-LSTM can better extract the spatial and time series feature information of EEG feature map. In this paper, the parameters of the FPN-LSTM model were set as shown in Table 2.

## 3. Experiments and Results

Emotion Analysis using Physiological singles (DEAP) dataset was used as the emotion recognition dataset, which was the EEG signals of 32 healthy subjects (16 males and 16 females). The dataset form is shown in Table 3.

### 3.1. EEG Feature Map Analysis

In this paper, the average power, variance power, and standard deviation power of 32 subjects’ EEG were extracted as the feature information of the EEG feature map. In this experiment, the EEG feature maps presented by the 2nd (male) and 24th (female) subjects under the same stimulus (10 experiments which were shown as the numbers 1–10 in Figure 4) were extracted and analyzed. The EEG feature maps of both subjects are shown in Figure 4.

It can be seen from Figure 4 that the EEG feature maps generated with average power as the feature sample had better performance than the EEG feature maps generated with variance power and standard deviation power as the feature sample; specifically, the edge contour of the EEG feature map was clearer, the pixel blocks were evenly distributed and brightly colored, and the EEG feature maps generated with the variance power as the feature were relatively weak in the EEG feature maps generated with the three features.

There were obvious differences in the color presentation and distribution of the EEG feature maps in Figure 4a,b, which indicated that men and women showed different personal emotions even when stimulated by the same stimulus source, which also fully showed the characteristics of individual differences in emotions.

EEG feature maps generated based on average power, standard deviation power, and variance power were used for emotion recognition, and the influence of the above three features on emotion recognition accuracy was analyzed. The feature maps based on the above three features were separately used as classification data for detection, and the emotion recognition accuracy is shown in Table 4.

It can be seen from Table 4 that the EEG feature maps generated based on the average power had achieved good recognition accuracy in terms of valence-arousal, which was 89.98% and 90.23%, respectively. It was 7.42% and 18.62% higher than standard deviation power and variance power in Valence state, and 6.44% and 17.89% higher than the Arousal state, respectively. It showed that the emotion classification results of EEG feature maps of average power can obtain better results.

### 3.2. EEG Channel Analysis

In order to improve the recognition accuracy, we proposed to strengthen the EEG channel with stronger emotional correlation and weaken the EEG channel with a weaker correlation. Therefore, the emotional correlation of the EEG channel data was tested. Among the 32 EEG channels, AF3, F3, F7, FC5, and T7 had a strong correlation with emotion [19], so we discussed the influence of the weight proportion (*w*) of the above five channels on the recognition accuracy. Table 5 and Figure 5 showed the accuracy rate of emotion recognition obtained by binary classification when AF3, F3, F7, FC5, and T7 channels were set with different weight proportions.

As can be seen from Figure 5 and Table 5, when the weight value of the five channels was set to 0.8, the emotion recognition accuracy was the highest, the recognition rate of the Valence state reached 90.05%, and the Arousal state reached 90.84%. The recognition accuracy without the weight proportion was close to *w* = 0.90, but the recognition rate decreased when w increased gradually. Through data analysis, it can be seen that when the data of the other channels were excessively weakened, the effective features of these channels would be lost, which can lead to the reduction in recognition rate. Therefore, we chose *w* = 0.80 as the weight proportion.

We extracted the two groups of poor EEG feature maps of two experiments and compared the original EEG feature map with the new EEG feature map obtained by applying *w* = 0.80, as shown in Figure 6. 

It can be seen from Figure 6 that the original EEG features map had lower contrast and changed to higher contrast after setting *w* = 0.8, especially related to emotional EEG channel position—its pixel brightness had improved, and the rest of the weakened EEG channel position pixels did not show much change. Based on the above situation, it can be seen that the enhanced channel effectively improves the proportion of effective features, and better weakens the invalid features, so as to effectively improve the accuracy of emotion recognition.

### 3.3. Analysis of FPN-LSTM Emotion Recognition Results

The FPN-LSTM and CNN-LSTM models were used to test the EEG data of 32 subjects in a DEAP EEG emotion database, and the Valence–Arousal emotion binary classification was carried out. The experiment was carried out in 200 iterations to obtain the accuracy and loss function data of the training set and test set, as shown in Figure 7 and Figure 8.

Figure 7 and Figure 8 showed the training and test results of the FPN-LSTM and CNN-LSTM model. As can be seen from the figures, during the training process, the accuracy of both training set and test set gradually improved, while the loss function gradually decreased; FPN-LSTM and CNN-LSTM converged basically after 100 iterations. It can be seen from Figure 7 that the accuracy of the CNN-LSTM model in the Valence state was stable at 0.95–1 for the training set and at 0.90 for the test set (Figure 7a); in the Arousal state, the training set was stable at near 0.95, and the test set was stable at 0.85–0.90 (Figure 7c). The accuracy of the FPN-LSTM model in the Valence state for the training set and the test set was higher than that of CNN-LSTM (Figure 8a); in the Arousal state, the accuracy of the training set and the test set were higher than that of CNN-LSTM (Figure 8c). The losses of FPN-LSTM in the Valence state and Arousal state were smaller than that of CNN-LSTM, which can be seen from Figure 7b,d and Figure 8b,d.

Based on the above data, we drew the confusion matrix of CNN-LSTM and FPN-LSTM, and the confusion matrix was 2 × 2. The emotion classification corresponding to Valence was high valence–low valence, and the emotion classification corresponding to Arousal was high arousal–low arousal, as shown in Figure 9. 

It can be seen from Figure 9a,b that the recognition accuracy of high valence, low valence, high arousal, and low arousal emotions using CNN-LSTM was higher than 85%, of which the accuracy of low valence recognition was the highest (91.66%) and the accuracy of high valence recognition was the lowest (85.86%); it can be seen from Figure 9c,d that the recognition accuracy of high valence, low valence, high arousal, and low arousal emotions using FPN-LSTM was higher than 87%, of which the recognition accuracy of high arousal was the highest (93.27%) and that of high valence was the lowest (87.69%). By comprehensive comparison of the two models, FPN-LSTM had an average increase of 1.29% in Valence and 2.20% in Arousal compared with CNN-LSTM. It showed that the classification accuracy is significantly improved by using FPN-LSTM. 

## 4. Discussions

Based on the above experimental results, in order to further analyze the performance of the method proposed in this paper, we compared the method proposed in this paper with the previous studies that used a DEAP dataset. Table 6 showed the comparison results between the method proposed in this paper and previous research methods.

It can be seen from Table 6 that the recognition accuracy of the method proposed in this paper in the Arousal state was higher than that of other literature in the table, and the recognition accuracy in the Valence state also had great advantages, among which the recognition accuracy of reference [15] in the Valence state was 0.57% higher than that of the proposed method in this paper; the reason for this is the method in this paper weakened the data of 27 channels when setting the weight proportion of EEG channels, and this part of data had a certain influence on the Valence state, so the recognition rate of the Valence state was slightly lower than the reference [15]; however, in the Arousal state, our method was 4.71% higher than reference [15]. The data used in reference [20], reference [21], reference [22], reference [23] and reference [24] were all 1D EEG data; in this paper, an EEG feature map was generated and used as the input. The accuracy of emotion recognition in the Valence and Arousal states were higher than the above four references.

In addition, we discussed the recognition accuracy of Valence and Arousal. It can be seen that in reference [15], reference [21], reference [22] and reference [23], the recognition accuracy of Valence was higher than that of Arousal, while in reference [20], reference [24] and the method proposed in this paper, the recognition accuracy of Arousal was higher than that of Valence. For this problem, we consider the following possibilities: (1) The influence of other physiological signals such as electrocardiogram (EOG) and electromyogram (EMG). In some studies, multi-source signals such as EEG, EOG, and EMG were used for emotion recognition, and different signals had different representations of different emotions, which further affected the recognition rate of the two emotions—namely, Valence and Arousal; (2) Different labeling methods had influence on the recognition accuracy of valence and arousal; (3) Channel selection was different. In order to improve the efficiency, many researchers chose to use part of the channels, which were in high correlation with emotion. This had influence on the recognition rate of Valence and Arousal. In this paper, emotion correlation analysis was carried out on EEG channels. High weight was assigned to the channels with high emotion correlation and low weight was assigned to the channels with low emotion correlation. This may have influenced the recognition rate of Valence and Arousal.

## 5. Conclusions

In this paper, it can be seen that the AEP and BiCubic interpolation method effectively retained the spatial topology information of EEG signals; the weight proportion distribution method for strong and weak channels of emotional correlation was proposed to improve the weight of emotional features of strong correlation channels; the FPN-LSTM model is proposed for emotion recognition, which can better integrate the features of different scales in the EEG feature maps. In conclusion, this research has demonstrated that our method achieved significantly better recognition accuracy and had good applicability in the field of EEG emotion recognition.

## Figures and Tables

**Figure 1 sensors-23-01622-f001:**
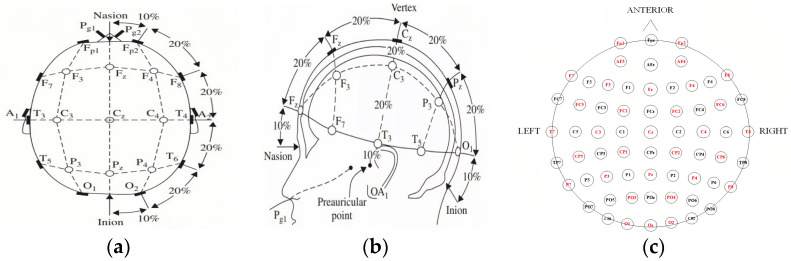
Electrode distribution diagram [18]. (**a**) Vertex electrodes distribution diagram; (**b**) Cranial electrodes distribution diagram; (**c**) 64-channel electrode distribution map of the International 10–20 System.

**Figure 2 sensors-23-01622-f002:**
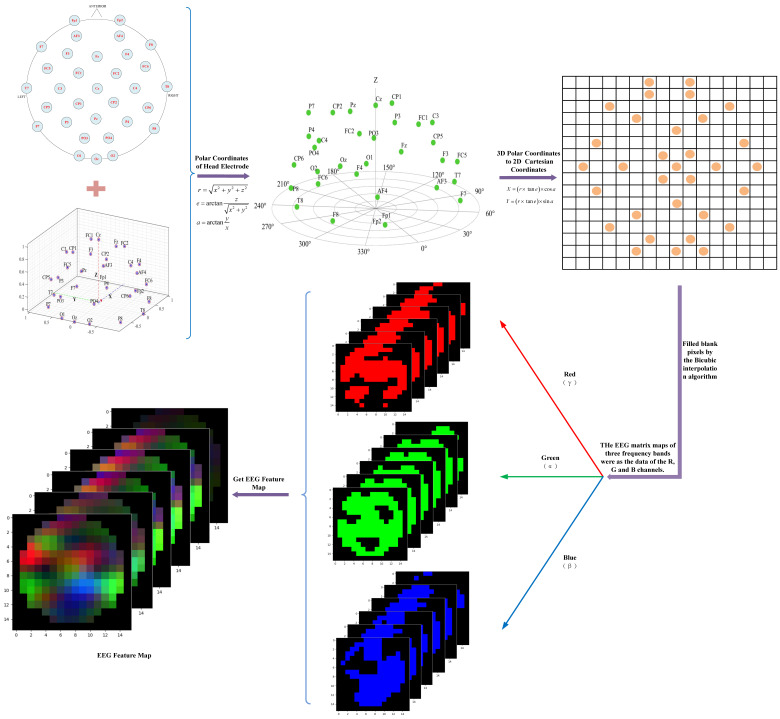
Schematic of EEG feature map generation.

**Figure 3 sensors-23-01622-f003:**
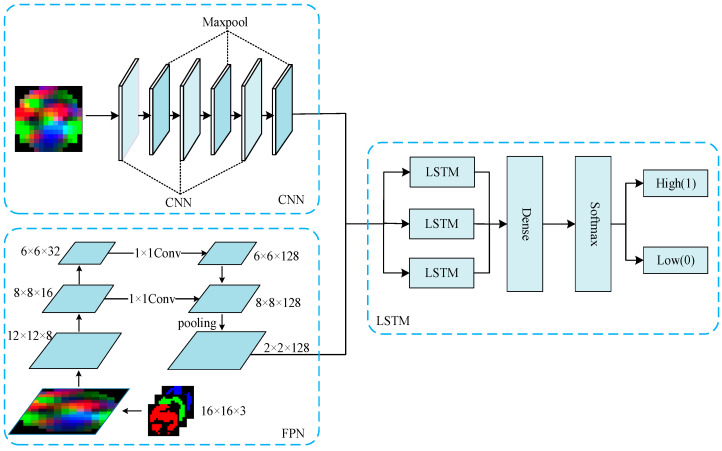
Model Structure Diagram of CNN-LSTM and FPN-LSTM.

**Figure 4 sensors-23-01622-f004:**
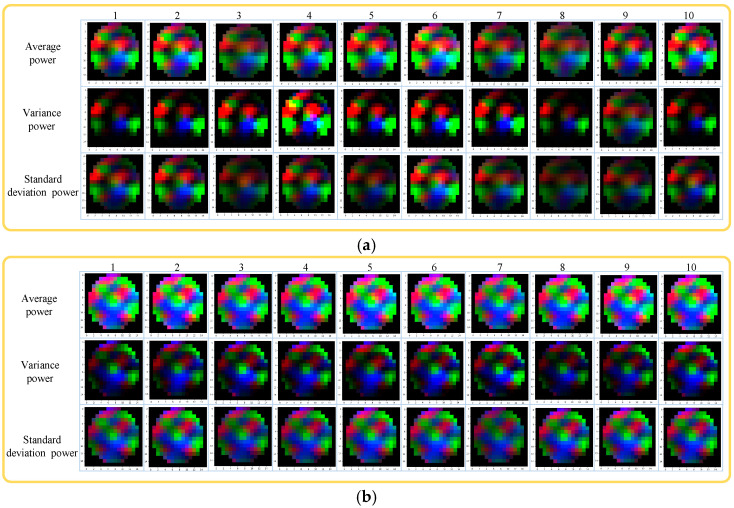
EEG feature maps (Average power, Variance power, and Standard deviation power). (**a**) EEG feature maps of the 2nd subject (male); (**b**) EEG feature maps of the 24th subject (female).

**Figure 5 sensors-23-01622-f005:**
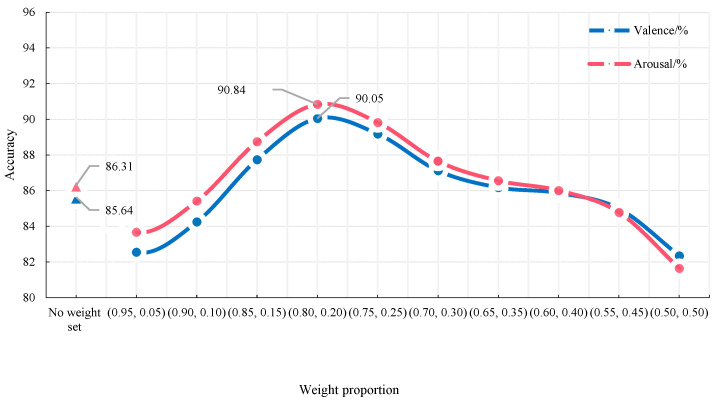
Comparison of EEG channel weight value.

**Figure 6 sensors-23-01622-f006:**
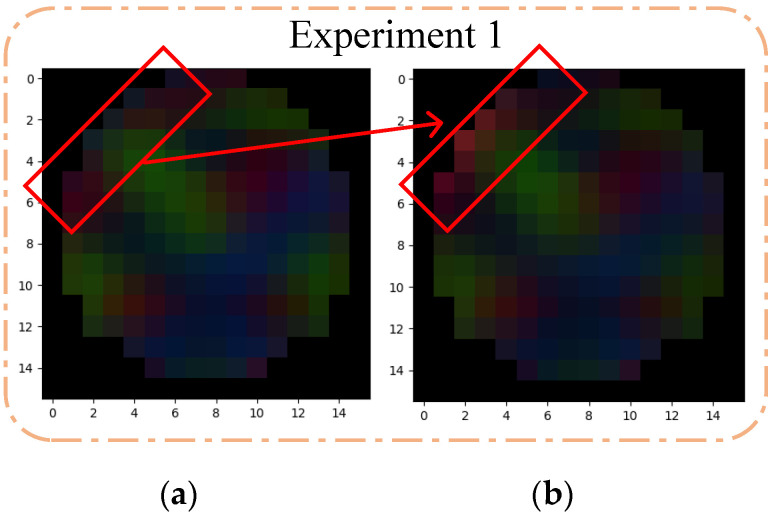
Comparison of the EEG features. (**a**) Original EEG feature map of experiment 1; (**b**) EEG feature map at *w* = 0.80 of experiment 1; (**c**) Original EEG feature map of experiment 2; (**d**) EEG feature map at *w* = 0.80 of experiment 2.

**Figure 7 sensors-23-01622-f007:**
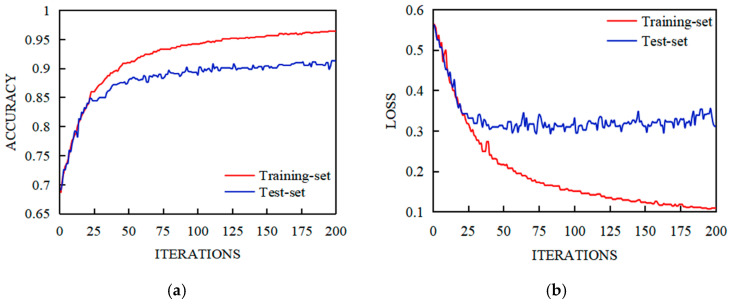
Training and test results of CNN-LSTM. (**a**) Accuracy of Valence Classification; (**b**) Loss of Valence Classification; (**c**) Accuracy of Arousal Classification; (**d**) Loss of Arousal Classification.

**Figure 8 sensors-23-01622-f008:**
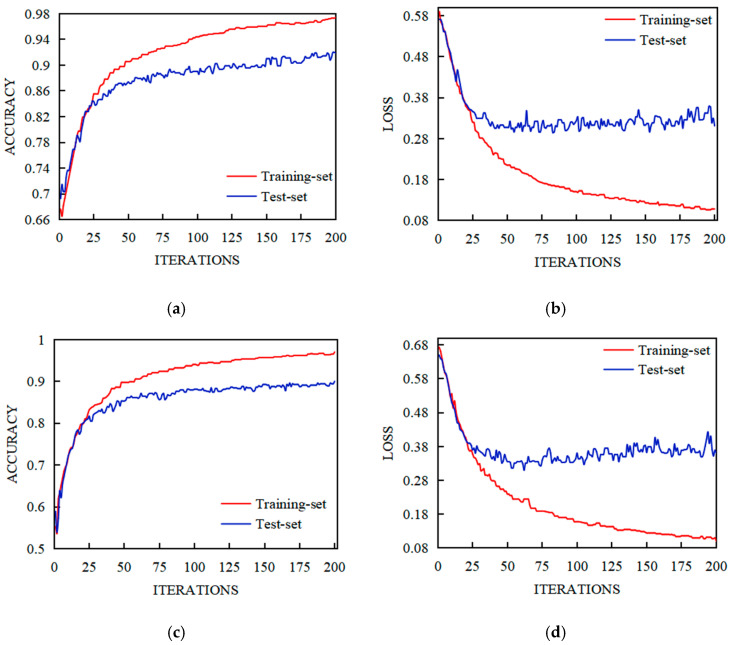
Training and test results of FPN-LSTM. (**a**) Accuracy of Valence Classification; (**b**) Loss of Valence Classification; (**c**) Accuracy of Arousal Classification; (**d**) Loss of Arousal Classification.

**Figure 9 sensors-23-01622-f009:**
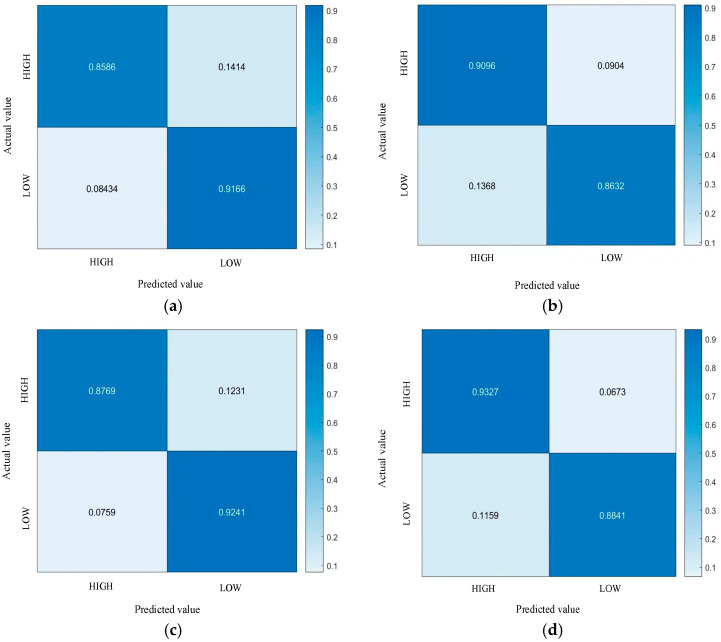
Confusion matrices of CNN-LSTM and FPN-LSTM. (**a**) The Valence of CNN-LSTM; (**b**) The Arousal of CNN-LSTM; (**c**) The Valence of FPN-LSTM; (**d**) The Arousal of FPN-LSTM.

**Table 1 sensors-23-01622-t001:** Representative characteristics of five basic rhythms.

Rhythm		Frequency (Hz)	Representative Characteristics
δ		0~4	It is commonly seen in the EEG of infants with brain hypoplasia, and in the deep sleep of adults with some brain diseases.
θ		5~8	It is often found in the state of exhaustion and deep thinking.
α	Slow	9~10	Before going to sleep, consciousness gradually moves towards a fuzzy brain state.
	Medium	10~11	Relaxed and focused. The body is in a comfortable state, and the mind is particularly active and can always inspire.
	Fast	12~13	In a state of high concentration and alertness.
β	Slow	14~16	In a state of concentration and ease.
	Medium	16.5~20	In the state of receiving various external information and thinking.
	Fast	20.5~30	In a state of agitation or excitement.
γ		≥31	In a state of happiness, stress relief, or thought.

**Table 2 sensors-23-01622-t002:** FPN-LSTM Parameters Setting.

Layer (Type)	Kernel Size	Number of Convolutional Kernels	Number of Parameters
Conv1	5 × 5	8	896
Conv2	5 × 5	16	18,496
Conv3	3 × 3	32	73,856
Pool	5 × 5	128	0
LSTM1	-	128	68,096
LSTM2	-	128	131,584
Desne1	-	100	1,683,850
Dense2	-	2	202

**Table 3 sensors-23-01622-t003:** The Data Form of DEAP.

Data Type	Data Size	Data Form
data	40 × 40 × 8064	video/trial × channel × data
labels	40 × 4	video/trial × label

**Table 4 sensors-23-01622-t004:** Recognition accuracy of different features.

Classification	Feature Type		
	Average Power/%	Standard Deviation Power/%	Variance Power/%
Valence	89.98	82.56	71.36
Arousal	90.23	83.79	72.34

**Table 5 sensors-23-01622-t005:** Recognition accuracy of different weight proportion.

Weight Proportion (*w,* 1 *− w*)	Accuracy	
	Valence/%	Arousal/%
No weight set	85.64	86.31
(0.95, 0.05)	82.55	83.67
(0.90, 0.10)	84.25	85.42
(0.85, 0.15)	87.74	88.75
(0.80, 0.20)	90.05	90.84
(0.75, 0.25)	89.18	89.82
(0.70, 0.30)	87.12	87.65
(0.65, 0.35)	86.18	86.56
(0.60, 0.40)	85.87	86.01
(0.55, 0.45)	84.99	84.78
(0.50, 0.50)	82.35	81.65

**Table 6 sensors-23-01622-t006:** Comparison of the proposed method with previous studies.

Study	Method	Classification	Test Accuracy (%)
Reference [20] (2017)	LSTM 1D EEG	Arousal	85.65
		Valence	85.45
Reference [21] (2018)	CNN 1D EEG + GSR	Arousal	76.56
		Valence	80.46
Reference [22] (2019)	LSTM − RNN 1D EEG	Arousal	74.38
		Valence	81.10
Reference [23] (2020)	KNN 1D EEG	Arousal	85.00
		Valence	86.30
Reference [15] (2020)	CNN − LSTM 2D EFM	Arousal	86.13
		Valence	90.62
Reference [24] (2022)	LSTM CNN 1D EEG	Arousal	69.50
		Valence	65.90
Proposed method	FPN − LSTM 2D EFM	Arousal	90.84
		Valence	90.05

## Data Availability

http://www.eecs.qmul.ac.uk/mmv/datasets/deap/download.html (accessed on 27 January 2023).

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
