# Peer review of "Feature Pyramid Networks and Long Short-Term Memory for EEG Feature Map-Based Emotion Recognition"

_sensors, 2023, doi:10.3390/s23031622_

Round 1

Reviewer 1 Report

The manuscript presents a novel emotion recognition based on EEG feature map and FPN-LSTM. The authors show the theoretical development along with a number of examples to demonstrate the efficiency of the methodology. The motivation of the manuscript is clearly stated and is also supported by interesting results. I enjoyed reading the paper and hope that my comments improve its quality.

1、Line82: Whether (0-4Hz) and (5-8Hz) cause impacts on the emotion recognition result?

2、Line 200: “Figure 4 indicated that men and women showed different personal emotions even when stimulated by the same stimulus source” . It should be described the reasons caused the differences.

Reviewer 2 Report

This paper proposed FPN-LSTM for emotion recognition and evaluated the performance.
However, this paper is unworthy to be published because of following reasons. 

  1. Figure 1 is very similar to the image used in “EEG Based Cognitive Workload Assessment for Maximum Efficiency”. Other stolen images and data may have been used.
  2. In abstract, authors used “so”. This word is not suitable for journal papers.
  3. Authors should use either α or alpha.
  4. The result said the classification accuracy for arousal was higher than that of valence. In contrast, valence is much more easily classified than arousal in previous studies. Authors should add discussion for this point.
  5. There is no discussion section in this paper.

Reviewer 3 Report

This work focuses on EEG Feature Map-based Emotion Recognition by proposing a new approach based on FPN and LSTM. Experimental results show superior performance of the proposed approach. The proposed approach is important for practitioners but this paper requires some modifications before the possible publication. Followings are my concerns:

1. Please highlight your contributions in introduction.

2. Compared with other existing approaches, what are the advantages of the proposed approach?

3. More descriptions are needed for Figure 3 since it is the most crucial part of the work.

4. The authors should consider more recent research done in the field of their study. For example, EEG-based emotion recognition using hybrid CNN and LSTM classification.

5. Is it sufficient to carry out 200 iterations for the proposed approach in the experiment?
